# Poloxamer-188 as a wetting agent for microfluidic resistive pulse sensing measurements of extracellular vesicles

**Mona Shahsavari**[1,2,3,4,5] *, **Rienk Nieuwland**[1,2,5,6], **Ton G. van Leeuwen**[1,3,4,5], **Edwin van der Pol**[1,2,3,4,5]

1 Amsterdam Vesicle Center, Amsterdam UMC, University of Amsterdam, Amsterdam, The Netherlands, 2 Laboratory of Experimental Clinical Chemistry, Amsterdam UMC, University of Amsterdam, Amsterdam, The Netherlands, 3 Biomedical Engineering and Physics, Amsterdam UMC, University of Amsterdam, Amsterdam, The Netherlands, 4 Amsterdam Cardiovascular Sciences, Atherosclerosis and Ischemic Syndromes, Amsterdam, The Netherlands, 5 Cancer Center Amsterdam, Imaging and Biomarkers, Amsterdam, The Netherlands, 6 Amsterdam Cardiovascular Sciences, Microcirculation, Amsterdam, The Netherlands

* m.shahsavari@amsterdamumc.nl

**Data Availability Statement:** All relevant data can be found in the following Figshare repositories. Flow cytometry data can be found at: https://doi.org/10.6084/m9.figshare.23260886 https://doi.org/

## Abstract

### Introduction

Microfluidic resistive pulse sensing (MRPS) can determine the concentration and size distribution of extracellular vesicles (EVs) by measuring the electrical resistance of single EVs passing through a pore. To ensure that the sample flows through the pore, the sample needs to contain a wetting agent, such as bovine serum albumin (BSA). BSA leaves EVs intact but occasionally results in unstable MRPS measurements. Here, we aim to find a new wetting agent by evaluating Poloxamer-188 and Tween-20.

### Methods

An EV test sample was prepared using an outdated erythrocyte blood bank concentrate. The EV test sample was diluted in Dulbecco's phosphate-buffered saline (DPBS) or DPBS containing 0.10% BSA (w/v), 0.050% Poloxamer-188 (v/v) or 1.00% Tween-20 (v/v). The effect of the wetting agents on the concentration and size distribution of EVs was determined by flow cytometry. To evaluate the precision of sample volume determination with MRPS, the interquartile range (IQR) of the particles transit time through the pore was examined. To validate that DPBS containing Poloxamer-188 yields reliable MRPS measurements, the repeatability of MRPS in measuring blood plasma samples was examined.

### Results

Flow cytometry results show that the size distribution of EVs in Tween 20, in contrast to Poloxamer-188, differs from the control measurements (DPBS and DPBS containing BSA). MRPS results show that Poloxamer-188 improves the precision of sample volume determination compared to BSA and Tween-20, because the IQR of the transit time of EVs in the test sample is 11 $\mu s$, which is lower than 56 $\mu s$ for BSA and 16 $\mu s$ for Tween-20.

10.6084/m9.figshare.23261090 Microfluidics resistive pulse sensing data can be found at: https://doi.org/10.6084/m9.figshare.23559945 The minimal data set underlying the results can be found at https://doi.org/10.6084/m9.figshare.24298507.

**Funding:** This work is part of the research program Perspectief with project number P18-26, which is financed by the Dutch Research Council (NWO). EvdP acknowledges funding from NWO, VIDI 19724. The funders had no role in the study design, data collection and analysis, decision to publish, or preparation of the manuscript.

**Competing interests:** The authors report no conflicts of interest.

Furthermore, the IQR of the transit time of particles in blood samples with Poloxamer-188 are 14, 16, and 14 µs, which confirms the reliability of MRPS measurements.

## Conclusion

The solution of 0.050% Poloxamer-188 in DPBS does not lyse EVs and results in repeatable and unimpeded MRPS measurements.

## Introduction

Extracellular vesicles (EVs) are cell-derived particles contributing to coagulation, intracellular communication, and waste management [1, 2]. As the biochemical content, concentration, and function of EVs change with disease, characterization of EVs is essential to exploit them clinically [3, 4]. Microfluidic resistive pulse sensing (MRPS; nCS1; Spectradyne, California, USA) is a single particle detection method that can determine particle size distribution (PSD) and the concentration of EVs in suspension.

MRPS is based on the Coulter principle. To determine the size of particles, MRPS applies a voltage over a pore. When a particle passes through the pore, the electrical resistance of the pore changes due to the difference in resistivity between the particle and the ambient fluid. This change is observed as a pulse of the measured voltage. The amplitude of this pulse is proportional to the particle volume and hence to the particle size cubed [5, 6]. Throughout this manuscript, we use the term size to refer to the diameter of spherical particles.

To estimate the concentration of particles, MRPS counts the number of particles and derives the sample volume. To determine the sample volume, the width of the pulse is measured, which corresponds to the transit time of the particle through the pore. The inverse of the transit time is proportional to the flow rate and together with the measurement time of a sample provides information about the measured sample volume [6].

For MRPS to work correctly, the sample must wet the pre-detection filter, the electrodes and the pore of the microfluidic chip, as depicted in Fig 1 of the manuscript by Fraikin et al. [6]. However, the chip is made of polydimethylsiloxane (PDMS), which is a hydrophobic material with a water contact angle >100˚ [7]. The hydrophobicity of PDMS hinders MRPS measurements of EV samples. To reduce the surface tension of EV samples, Spectradyne recommends to use 1.00% Tween-20 (v/v) as a wetting agent. However, Tween-20 destabilizes the membrane and morphology of erythrocytes [8–12] and might therefore also affect the membrane of EVs. Cimorelli et al. [13] indeed confirmed that Tween-20 decreases the detectable concentration of EVs and therefore introduced bovine serum albumin (BSA) as an alternative. Although the effect of Tween-20 on stained EVs has been studied [13], it remains unclear whether Tween-20 causes EV lysis, and/or interferes with EV staining.

After two-years of evaluating the protocol of Cimorelli et al. in our lab, we found that MRPS measurement results using BSA as a wetting agent were not repeatable. Fig 1A shows the concentration and PSD in diluted human blood plasma, which was measured three times with MRPS containing 0.10% BSA in Dulbecco's phosphate-buffered saline (DPBS). The total concentration and PSD measured during the second run differed from the other measurements, demonstrating poor repeatability.

The stability of the flow rate, which is reflected in the distribution of the transit time of the particles, is influenced by the diluted wetting agent in DPBS. Flow rate instability can result in an imprecise estimation of the measured sample volume, leading to deviations in the

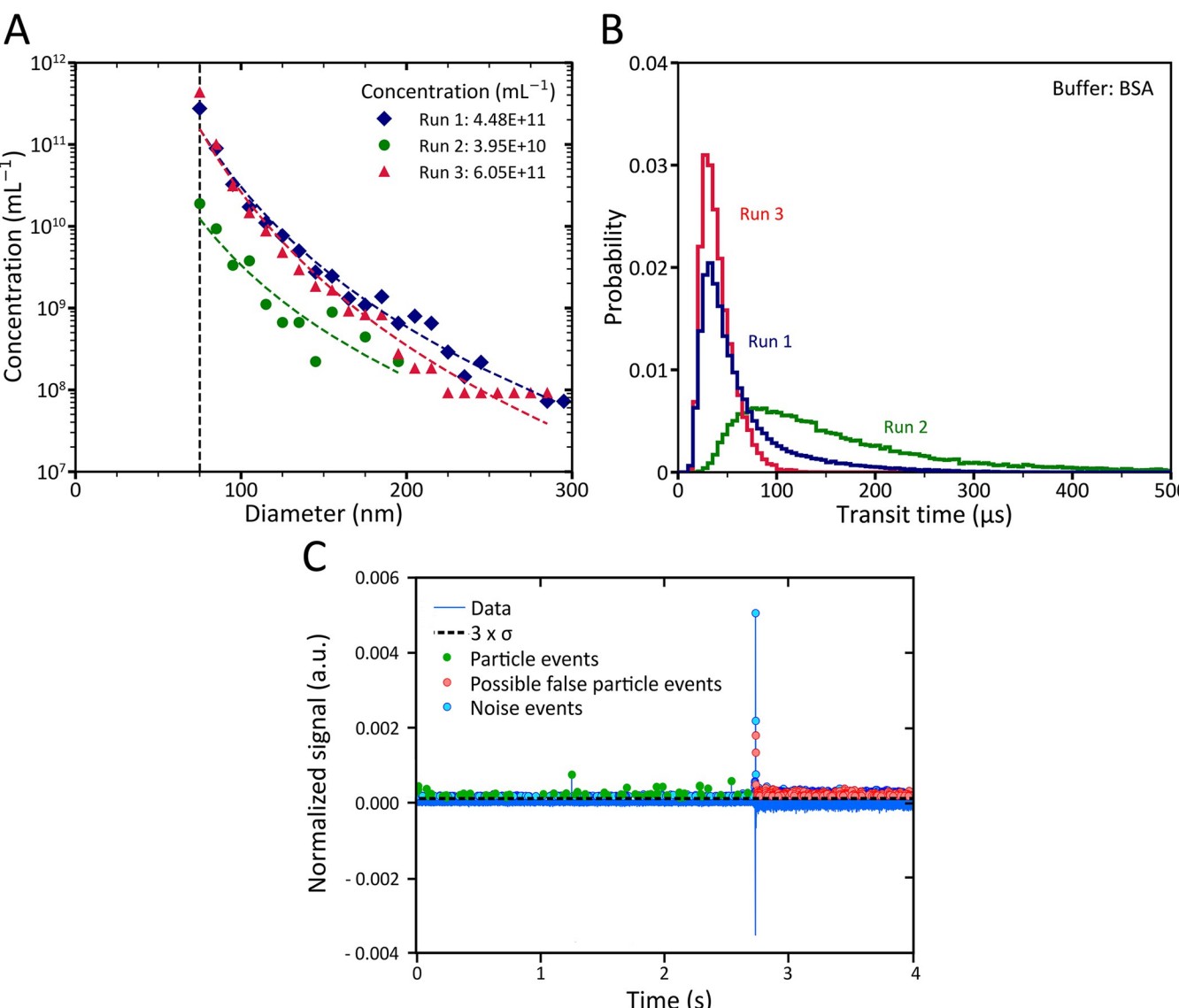

**Fig 1. Microfluidic resistive pulse sensing (MRPS) data of a human blood plasma sample that is 150-fold diluted in Dulbecco's phosphate-buffered saline (DPBS) containing 0.10% bovine serum albumin (BSA).** (A) Diamond, circle, and triangle symbols represent the particle size distribution (PSD) of the plasma sample measured in 3 independent MRPS measurements with a C400 cartridge. Dashed lines visualize fitted power law functions on PSDs. The parameters and coefficient of determinations for the fitted power law functions are as follows: $a = 6.822 \times 10^{21}$, $k = 5.678$, and $R^2 = 0.98$ for run 1; $a = 3.901 \times 10^{18}$, $k = 4.534$, and $R^2 = 0.86$ for run 2; and $a = 7.108 \times 10^{22}$, $k = 6.219$, and $R^2 = 0.97$ for run 3. The vertical dashed line at 75 nm represents the smallest reported bin size for the MRPS measurements. (B) The distribution of particle transit time in three independent MRPS measurements with a bin width of 5 μs. (C) Normalized signal (a.u.) versus time for particles in human blood plasma measured in run 1. The horizontal dashed line represents the threshold of peak detection, green dots are peaks that are representative of particle events, blue dots are noise events, and red dots are incorrect particle events that were determined based on the manufacturer's peak filter recommendations.

determined particle concentration. Fig 1B shows the transit time distribution of the detected events with a signal to noise ratio (S/N) >10 for the measurements in Fig 1A. The interquartile range (IQR) of the transit time distribution is 41 μs for run 1, 155 μs for run 2, and 20 μs for run 3. As the sample volume is determined based on the average transit time, the larger IQR for run 2 indicates that the sample volume was less precisely determined compared to runs 1 and 3.

Fig 1C shows 4 seconds of the normalized raw signal recorded during run 1, which exhibits a sudden increase in the background noise. Inadequate wetting of the pore is expected to decrease the background signal while the noise of the normalized background signal increases. Fluctuations in the background noise can cause the peak differentiation algorithm to malfunction, resulting in imprecise concentration estimations, particularly for particles with signals close to the background noise.

This study aims to identify a wetting agent that lowers the surface tension of water to achieve optimal pore wetting, while exerting well-controlled effects on EVs. The hallmark of EVs is their phospholipid membrane [14], which require a surface tension to maintain structural integrity. However, the phospholipid membranes of EVs may contain cross-linking proteins that increases their rigidity in the presence of wetting agents [15]. Here we will explore the potential of two non-ionic wetting agents, Poloxamer-188 and Tween-20 for measuring EVs with MRPS.

To find the desired wetting agent, an EV test sample, containing mostly EVs, was prepared and used to explore the effect of Poloxamer-188 and Tween-20 on the concentration and PSD of unstained EVs using flow cytometry (FCM). FCM was selected over MRPS because FCM measurements less dependent on the used wetting agents. To investigate the possible effect of wetting agents on the precision of MRPS in sample volume determination, the transit time distribution of the diluted EV test sample in DPBS containing BSA, Poloxamer-188, or Tween-20 was examined. Eventually, to validate that DPBS with Poloxamer-188 yields reliable MRPS measurements of commonly used biofluids, the repeatability of MRPS in measuring EVs and other particles in human blood plasma samples was examined.

## Methods

### Biological sample preparation

**EV test sample.** To prepare the EV-containing test sample, an outdated erythrocyte blood bank concentrate was used as the starting material. After opening the blood bank bag, the sample was transferred into 10-mL conical centrifuge tubes without any anticoagulation. To remove erythrocytes, the sample was centrifuged twice at 2,500 g at 20°C for 15 minutes without brake (Rotina 46RS Hettich Zentrifugen, Tuttlingen, Germany). After each centrifugation step, the supernatant was pipetted down to 10-mm above the pellet and transferred into a new conical centrifuge tube. Aliquots of 500 μl of the supernatant were snap-frozen in liquid nitrogen, stored at −80°C, and thawed in a water bath at 37°C for 2 minutes before use.

Cluster of differentiation (CD) 235a-Phycoerythrine (PE; Agilent Dako, California, USA, clone JC159) was used to stain erythrocyte-derived EVs in the EV test sample. CD235a binds to Glycophorin A, which is a typical protein for erythrocytes. To remove antibody aggregates from the staining reagent, 4-fold diluted CD235a-PE in DPBS was centrifuged at 18,890x g for 5 min at 20°C (MIKRO 220R, Hettich Zentrifugen). To stain the EV test sample, 20 μL of the 24-fold diluted (in DPBS) test sample was incubated with 2.5 μL of diluted CD235a-PE for two hours. After incubation, samples were further diluted by adding 200 μL of DPBS or DPBS containing 0.10% BSA (w/v), 0.050% Poloxamer-188 (v/v), 0.10% Triton X-100 (v/v), or 1.00% Tween-20 (v/v).

**Blood plasma.** Blood was collected using 21G needles from 8 healthy donors (5 female and 3 male). The Amsterdam UMC Ethical Review Board, location AMC, has stated that collecting blood from healthy donors for preclinical research is exempt from the WMO (the Medical Research Involving Human Subject Act). Consequently, no additional approval from the board was required. All procedures involving human participants adhered to the Declaration of Helsinki, and verbal informed consent was obtained. After discarding the first 2 mL of

blood, blood was collected in two 6 mL BD Vacutainer blood collection tubes containing the anticoagulant EDTA. To reach an appropriate ratio between blood volume and anticoagulant, the collection tubes were filled till the indication. The blood was pooled and the pooled sample was centrifuged (Rotina 46RS Hettich Zentrifugen) in two subsequent centrifugation steps at 2,500 g at 20˚C for 15 minutes without brake. After each centrifugation step, the supernatant was collected to 10 mm above the bottom of the tube, and then transferred into a new plastic tube. 50 μL aliquots of cell-depleted plasma sample were snap-frozen in liquid nitrogen, stored at −80˚C, and thawed at 37˚C for 1 minute before use.

## Preparation of diluents

We prepared DPBS containing 0.10% BSA (w/v), 0.050% Poloxamer-188 (v/v), 0.10% Triton X-100 (v/v), and 1.00% Tween-20 (v/v) which are termed BSA, Poloxamer, Triton and Tween, respectively, throughout this manuscript. Diluents were filtered (Nuclepore Track-Etch Membrane, 47 mm, 0.05 μm, Whatman, Maidstone, UK) to remove particles larger than 50 nm. BSA was filtered using centrifugation filters (Vivaspin 500, 100 kDa filters, Sartorius, Göttingen, Germany) at 12,000 g for 5 minutes at room temperature. The concentrations of Poloxamer and Tween were determined based on the recommendation of Spectradyne.

## MRPS measurements

To perform and analyze MRPS measurements, nCS1 hardware (v0, Spectradyne) and nCS1 viewer software (version 2.5.0.297, Spectradyne) were used. To differentiate signals of particles from the noise, the applied peak filters were chosen based on the recommendations of the manufacturer (Signal/noise [S/N] >10, size >65 nm, transit time <60 μs or <80 μs based on Mold-ID of the cartridge, 0.2< peak symmetry index <4.0). The nCS1 viewer software assumes that the measured sample volume is inversely related to the average transit time of particles passing through the pore, and assesses the sample volume through a calibration. We used C400 cartridges with a specified detection range from 65 nm to 400 nm. A bin size of 10 nm was chosen for representing measured PSD with MRPS. The first bin of the measured PSD with MRPS, 65 nm, shows the concentration of particles with a size range between 60 to 70 nm. As the applied size filter removes particles with a size below 65 nm, the first data point of the PSD does not accurately represent the concentration and is therefore not shown. The reported total concentration comprises particles within a size range of 70 to 400 nm. All the measured PSDs with MRPS were fitted with power law functions:

$$C = a \times d^{-k} \tag{1}$$

where C is concentration, d is diameter, with a and k are fit parameters. To obtain these fit parameters, a linear regression was applied on log transformed data of both the diameter and concentration.

## Flow cytometry measurements

Flow cytometry measurements were performed using an Apogee A60-Micro. The applied trigger threshold on side scattering was corresponding to ~145 nm in terms of the optical size for EVs. Samples were measured for 120 seconds at a flow rate of 3.01 μl/min. Samples were diluted such that the resulting count rates were <5000 /s to prevent swarm detection [16]. All details about the FCM experiments can be found in the completed MIFlowCyt-EV template [14] (Supplemental materials).

Rosetta Calibration (v1.24, Exometry, the Netherlands) was used to derive the size of measured particles from the calibrated light scattering intensity. We assumed that all particles in the EV test sample were EVs and that EVs had a core refractive index (RI) of 1.38, a shell RI of 1.48 and a shell thickness of 6 nm [17]. Calibration of fluorescence signals was done using SPHERO™ PE MESF beads (AK01, Spherotech Inc., Irma Lee Circle, IL, USA). FCM data were processed using FlowJo (v10, BD, CA, USA) and custom-developed software (MATLAB R2020b, MathWorks, USA).

Erythrocyte-derived EVs were stained with CD235a-PE and measured with FCM. As a negative control, to show how large the lysed EV fraction is compared to the total particle concentration, the unstained EV test sample was diluted in Triton and measured with FCM. In addition, to confirm that the majority of particles in the test sample were EVs, the flow cytometry scatter ratio (Flow-SR) was used to show that within the applicable range of Flow-SR particles have an RI <1.42 [3]. The details of the applicable range of Flow-SR can be found in the completed MIFlowCyt-EV template (supplemental materials).

To investigate the effect of the wetting agents on the concentration and PSD of EVs, the unstained EV test sample was diluted in filtered DPBS, BSA, Poloxamer, or Tween and subsequently analyzed using FCM. Here, an unstained EV test sample was used to account for the potential interference of wetting agents with the stainstaining process of EVs.

## Data processing

Data processing and preparation of the graphs were done in Python (v3.8.5).

## Results

### EV test sample characterization

To evaluate whether EVs remain intact in the presence of Poloxamer and Tween, we developed an EV-containing test sample. To measure which fraction of all particles present in the EV test sample are EVs, FCM was used, because FCM can identify stained EVs from unstained particles while MRPS cannot. Fig 2A shows the PSD of all particles and EVs stained with CD235a in the EV test sample measured with FCM. The detection range of the used scatter detector corresponds to EVs with a size between 145 nm and 800 nm. The total concentration of detected particles in the EV test sample is 2.0E10 $ml^{-1}$. After the addition of Triton, the measured concentration decreased to 7.1E8 $ml^{-1}$, which is similar to the concentration of particles measured in Triton (supplemental materials). This observation implies that 97% of all particles were susceptible to detergent lysis, indicating that most particles >145 nm are EVs. In addition, in total 68% of the particles >145 nm and 96% of the particles >285 nm in the EV test sample are stained with CD235a, and therefore likely EVs.

In Fig 2B, we highlight the impact of the fluorescence gate at 118 molecules of equivalent soluble fluorophores (MESF), which not only excludes background noise but also stained events with a fluorescence intensity below 118 MESF. Notably, the data points with the highest density (shown in yellow) and a fluorescence intensity of 118 MESF correspond to particles with an upper size of approximately 285 nm. Consequently, for sizes <285 nm, the fluorescence gate results in a reduced percentage of measured stained particles compared to the total number of particles. This observation shows that the actual percentage of CD235a-PE+ particles exceeds 68%, confirming that the majority of particles >145 nm in the test sample are indeed EVs.

Fig 2C shows that for 89% of the particles in the test sample with a size between 200–650 nm, which is the applicable range of Flow-SR [4], the RI is <1.42. This RI is typical for EVs within this size range and thereby further confirms that the majority of measured particles in

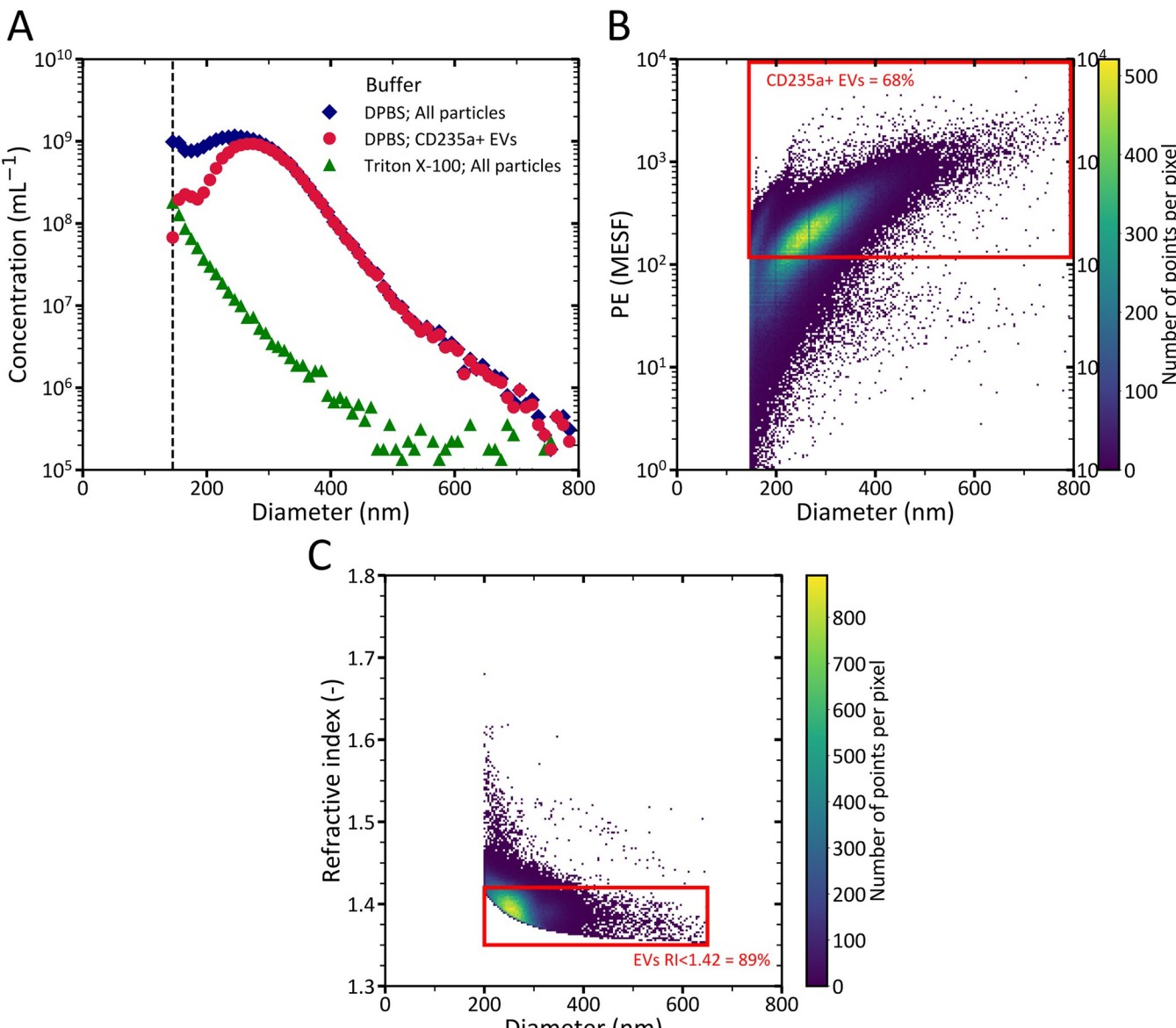

**Fig 2. Flow cytometry (FCM) data of the extracellular vesicle (EV) test sample.** (A) Concentration versus size of particles. The vertical dashed line represents the trigger threshold of FCM, which is 145 nm. Diamonds represent the particle size distribution (PSD) of all particles in the EV test sample, circles represent the PSD of EVs stained with cluster of differentiation (CD) 235a, and triangles represent the PSD of remaining particles in the EV test sample after detergent lysis of EVs with 0.10% Triton X-100 in Dulbecco's phosphate-buffered saline (DPBS). To relate scatter to size, EVs were modelled as particles with a core refractive index (RI) of 1.38, a shell RI of 1.48 and a shell thickness of 6 nm. (B) Fluorescence intensity of phycoerythrin (PE) versus size of particles measured in the EV test sample. The fluorescent gate of 118 molecules of equivalent soluble fluorochrome (MESF), which was based on an unstained control, was exceeded by 68% of particles. (C) Refractive index versus size of measured particles with FCM in the applicable range of the flow cytometry scatter ratio (Flow-SR; 47% of all detected particles in the EV test sample are within the applicable range of Flow-SR), among which 89% of particles have a RI <1.42.

the developed test sample by FCM are EVs [4, 18, 19]. Based on the findings in Fig 2, we assume that all particles with sizes >145 nm in the test sample are EVs.

### Effect of wetting agents on EVs

To evaluate whether EVs remain intact in the presence of Poloxamer and Tween, Fig 3 shows the measured concentration of the unstained EV test sample diluted in DPBS, BSA,

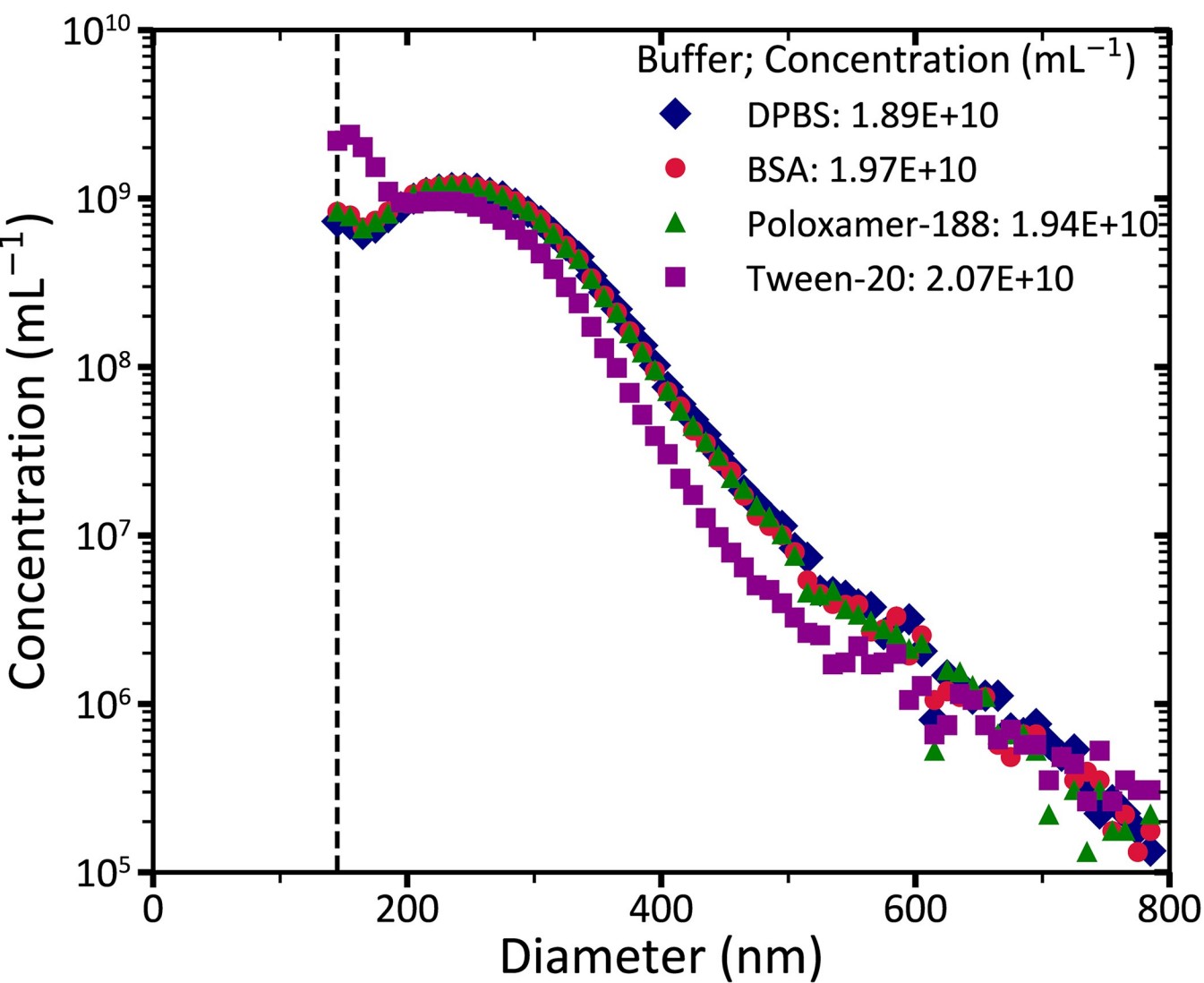

**Fig 3. Concentration versus the size of particles in the extracellular vesicle (EV) test sample measured with flow cytometry (FCM).** The vertical dashed line represents the trigger threshold of FCM, which is 145 nm. Diamonds, circles, triangles, and squares represent the particle size distribution of the 264-fold diluted EV test samples in Dulbecco's phosphate-buffered saline (DPBS) or DPBS containing 0.10% BSA (w/v), 0.050% Poloxamer-188 (v/v), or 1.00% Tween-20 (v/v), respectively. To relate scatter to size, EVs were modelled as particles with a core refractive index of 1.38, a shell refractive index of 1.48 and a shell thickness of 6 nm.

Poloxamer, and Tween using FCM. Whereas the total concentration of EVs is unaffected in the presence of Poloxamer and Tween compared to the controls (DPBS and BSA), the PSD of EVs diluted in Tween differs substantially from the controls (DPBS and BSA). Compared to the controls, the concentration of EVs <235 nm increases by 54% and of EVs >235 nm decreases by 34% when diluted in Tween. In sum, the PSD of EVs are affected by Tween, but remain intact in the presence of Poloxamer.

### MRPS compatibility of explored wetting agents

To study the compatibility of wetting agents with MRPS measurements of EVs, Fig 4A shows the transit time distribution of particles passing through the pore in three MRPS measurements of the EV test samples that were diluted in BSA, Poloxamer, and Tween. The IQR of the

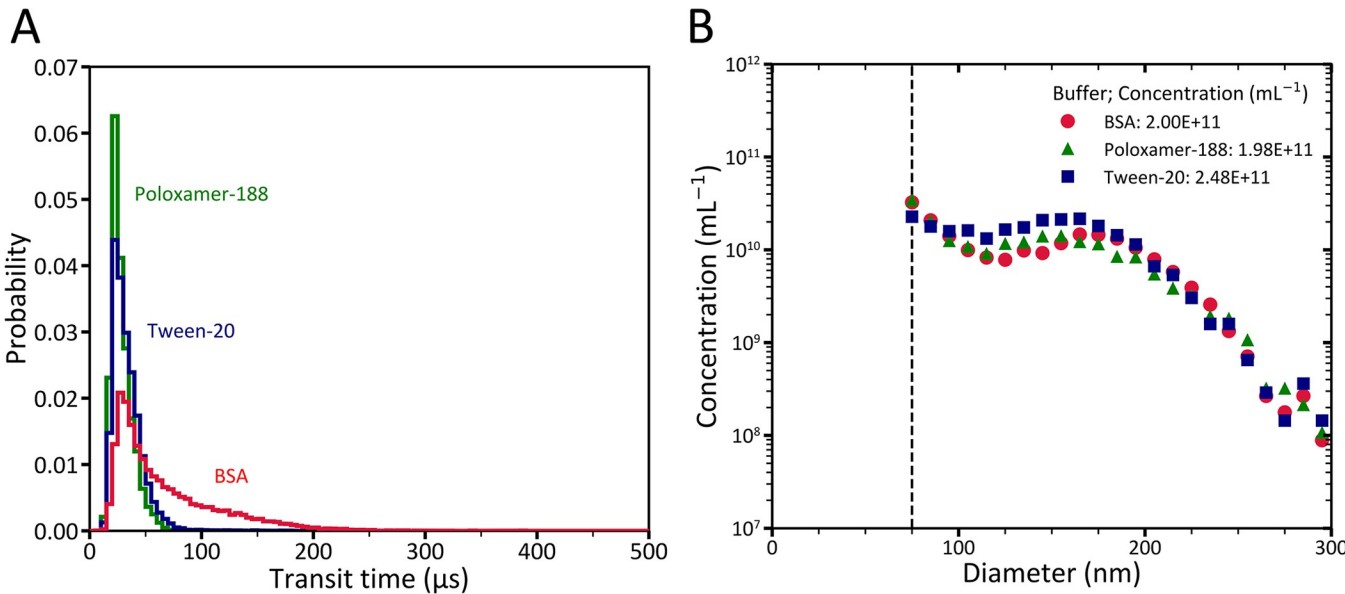

**Fig 4. Microfluidic resistive pulse sensing (MRPS) measurements of the extracellular vesicle (EV) test sample.** (A) Red, green and blue lines show the transit time distribution (5-μs bins) of events with a signal/noise >10 of the diluted EV test sample in Dulbecco's phosphate-buffered saline (DPBS) containing 0.10% bovine serum albumin (BSA) (w/v), 0.050% Poloxamer-188 (v/v), or 1.00% Tween-20 (v/v), respectively. The interquartile range of the transit time of particles through the pore is 11 μs for Poloxamer-188, 56 μs for BSA and 16 μs for Tween-20. (B) Concentration versus the size of all particles in the EV test sample, which was diluted in DPBS containing BSA (Triangle), Poloxamer-188 (diamond), or Tween-20 (circle). The dashed line shows the lower limit of detection for size measurements with MRPS.

transit time is 11 μs for Poloxamer, 56 μs for BSA and 16 μs for Tween. The lower IQR of the transit time of particles with Poloxamer, in comparison to BSA and Tween, indicates higher precision in determining the sample volume. Thus, the use of Poloxamer as a wetting agent yields the best results in pore wetting.

Fig 4B shows the concentration versus size of the EVs in the EV test sample diluted in DPBS containing BSA, Poloxamer, and Tween as measured by MRPS. The concentration of particles in the presence of Tween is 24% higher compared to BSA and Poloxamer. Within the size range of 145 nm to 300 nm, this result aligns with our FCM findings and confirms that Tween affects MRPS measurements of EVs.

### Evaluation of Poloxamer for MRPS measurements

To investigate the repeatability of MRPS measurements in measuring the concentration and PSD of commonly used biofluids, plasma samples diluted in poloxamer were measured three times with MRPS. Fig 5A shows that the experiment resulted in repeatable concentration and PSD measurements. Fig 5B shows the transit time distribution for the same measurements as in panel A. The IQRs of the transit times are 14, 16, and 14 μs, which are negligible differences compared to the variation when using BSA (Fig 1B). The results in Fig 5 confirm that Poloxamer provides reliable MRPS measurements.

### Discussion

Here, we aimed to improve the protocol developed by Cimorelli et al. [13] for MRPS measurements of EV-containing samples by exploring the use of non-ionic wetting agents as wetting agents. The desired wetting agent should fulfill two criteria. First, the wetting agent should minimally affect the EVs. Second, the wetting agent should result in pore wetting, which can

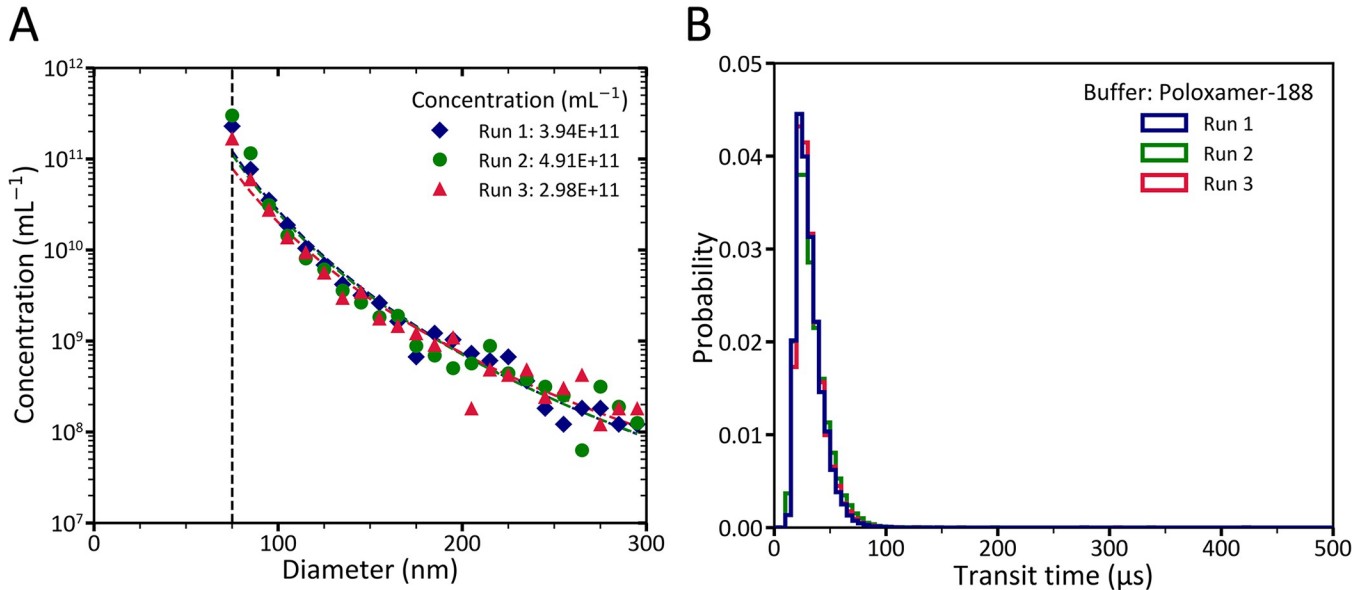

**Fig 5. Microfluidic resistive pulse sensing (MRPS) data of 150-fold diluted blood plasma in Dulbecco's phosphate-buffered saline (DPBS) containing 0.050% Poloxamer-188.** (A) Diamonds, circles, and triangles show the particle size distribution (PSD) of 3 independent MRPS measurements of the plasma sample fitted with power law functions. The determined parameters and coefficient of determinations for the fitted power law functions are as follows: a = 7.734× $10^{20}$, k = 5.228, and $R^2$ = 0.98 for run 1; a = 5.235× $10^{20}$, k = 5.158, and $R^2$ = 0.95 for run 2; and a = 6.911× $10^{19}$, k = 4.769, and $R^2$ = 0.95 for run 3. The vertical dashed line represents the smallest reported bin size in this study for the MRPS measurements using C400 cartridges. (B) The distribution of particles transit times in three MRPS measurements of the plasma sample with a bin width of 5 μs.

be indirectly observed by measuring the particle transit time. To study the effect of wetting agents on EVs, we developed an EV test sample based on an outdated erythrocyte blood bank concentrate. We used FCM to show that most particles in the test sample are EVs, because FCM, in contrast to MRPS, can differentiate EVs from other particles. FCM data show that within the detection range of 145 to 800 nm, at least 68% of the particles in this sample are EVs.

The protocol of Cimorelli et al. showed that the measured concentration of stained EVs in blood plasma decreases in the presence of Tween [13]. However, it was unclear whether Tween lysed EVs or interfered with the immunostaining. To clarify the effect of wetting agents on EV structure, we studied the effect of 0.050% Poloxamer (v/v) and 1.00% Tween (v/v) in DPBS on unstained EVs in an EV test sample. We chose FCM over MRPS because FCM measurements are less susceptible to wetting agents than MRPS and FCM also allows reliable size determination [20]. We did not observe any effect on the PSD and total concentration of EVs in the presence of Poloxamer. Therefore, we concluded that Poloxamer does not lyse EVs and thereby satisfies our first criterion for the desired wetting agent.

Although the total concentration of EVs in the presence of Tween remained almost unchanged (difference <7.3%, Fig 3), the size distribution of EVs deviated from the controls (DPBS and BSA). In the presence of Tween, the concentration of EVs <235 nm increases by 54% and of EVs >235 nm decreases by 34%, thereby resulting in a negligible change in the total concentration of EVs compared to the controls. Possible explanations for the deviated PSD are that Tween lyses relatively large EVs into smaller vesicles, or that Tween forms detectable micelles <235 nm. Regardless of the reason behind this deviation, we concluded that Tween affects the PSD and concentration of EVs. Our results further show that both the PSD and total concentration need to be considered when studying the stability of EV populations.

To evaluate the reliability of MRPS results in the presence of the examined wetting agents, we used the stability of the flow rate as a criterion. This criterion is inversely related to the width of the distribution of time that particles transit through the pore. The transit time distribution of EVs diluted in BSA, Poloxamer, and Tween showed that Poloxamer results in the most reliable MRPS data because the corresponding width of the transit time distribution is shortest compared to other wetting agents (Fig 4).

In sum, the solution of 0.050% Poloxamer-188 in DPBS satisfies both criteria of the desired wetting agent and can be used for MRPS measurements of EVs. We further evaluated the reliability of the developed protocol by measuring particles in human blood plasma sample three times, and the findings indicated repeatable results (Fig 5).

Recently MRPS extended with fluorescence detection (ARC, Spectradyne, California, USA) became commercially available. To check the applicability of the developed protocol for ARC measurements, we explored the effect of wetting agents on stained EVs with CD235a (supplemental materials). We observed no variation in concentration and PSD of stained EVs in the presence of Poloxamer. Even though this preliminary result shows the applicability of the developed protocol also for ARC measurements of stained EVs with CD235a, further investigation is required to check the effect of Poloxamer on stained EVs with other EV-specific biomarkers.

## Supporting information

**S1 Fig. Scatter calibrations.** (A) Forward scatter and (B) side scatter calibration of the A60-Micro by Rosetta Calibration. To relate scatter to the size of EVs, we modelled EVs as core-shell particles with a core refractive index of 1.38, a shell refractive index of 1.48, and a shell thickness of 6 nm.
(TIF)

**S2 Fig. Flow cytometry analysis of EV test sample.** (A) Concentration versus diameter of all particles (diamonds), EVs labeled with CD235a (circles), particles remaining after detergent lysis with 0.10% Triton X-100 in Dulbecco's phosphate-buffered saline (triangles). To relate scatter to diameter, EVs were modelled as particles with a shell refractive index of 1.48, a shell thickness of 6 nm, and a core refractive index of 1.38. The vertical dashed line marks the trigger threshold for all particles, set at 145 nm. (B) Refractive index versus diameter of particles within the applicable range of the flow cytometry scatter ratio (Flow-SR), which includes 47% of all events. Notably, 99% of these particles show a refractive index below 1.42.
(TIF)

**S3 Fig. Two independent measurements of the extracellular vesicle (EV) test sample with flow cytometry (FCM).** (A, B) Concentration versus diameter of 267- fold diluted EV tests sample in Dulbecco's phosphate-buffered saline (DPBS; diamond), and DPBS containing 0.10% BSA (w/v; circle), 0.050% Poloxamer-188 (v/v; triangle), and 1.00% Tween-20 (v/v; square). To establish a relationship between scatter and diameter, EVs were modeled as particles with a shell refractive index of 1.48, a shell thickness of 6 nm, and a core refractive index of 1.38. The vertical dashed line shows the trigger threshold of FCM, which is 145 nm.
(TIF)

**S4 Fig. Concentration of particles in a lysed extracellular vesicle (EV) test sample (diamonds) and in Triton X-100 (circles) versus diameter measured with flow cytometry (FCM).** Both size distributions are overlapping and show a steep decrease in concentration with increasing diameter.
(TIF)

**S5 Fig. Concentration versus the diameter of particles in the extracellular vesicle (EV) test sample measured with flow cytometry (FCM).** The vertical dashed line represents the trigger threshold of FCM, which is 145 nm. Diamonds, circles, triangles, and squares represent the particle size distribution of CD235a+ EVs in Dulbecco's phosphate-buffered saline (DPBS) or DPBS containing 0.10% BSA (w/v), 0.050% Poloxamer-188 (v/v), or 1.00% Tween-20 (v/v), respectively. To relate scatter to diameter, EVs were modelled as particles with a core refractive index of 1.38, a shell refractive index of 1.48 and a shell thickness of 6 nm.
(TIF)

**S1 File. The MIFlowCyt-EV documentation.**
(DOCX)

**S2 File. The MIFlowCyt-EV documentation.**
(DOCX)

**S3 File. Supporting data.**
(DOCX)

**S1 Data.**
(ZIP)

## Author Contributions

**Conceptualization:** Mona Shahsavari, Rienk Nieuwland, Ton G. van Leeuwen, Edwin van der Pol.

**Data curation:** Mona Shahsavari.

**Formal analysis:** Mona Shahsavari.

**Supervision:** Rienk Nieuwland, Ton G. van Leeuwen, Edwin van der Pol.

**Validation:** Mona Shahsavari, Rienk Nieuwland, Ton G. van Leeuwen, Edwin van der Pol.

**Visualization:** Mona Shahsavari.

**Writing – original draft:** Mona Shahsavari.

**Writing – review & editing:** Mona Shahsavari.

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
