## [Decision Letter · Decision Letter 0]

1 Sep 2023

PONE-D-23-19466Poloxamer-188 as a wetting agent for microfluidic resistive pulse sensing measurements of extracellular vesiclesPLOS ONE

Dear Dr. Shahsavari,

Thank you for submitting your manuscript to PLOS ONE. After careful consideration, we feel that it has merit but does not fully meet PLOS ONE’s publication criteria as it currently stands. Therefore, we invite you to submit a revised version of the manuscript that addresses the points raised during the review process.

We look forward to receiving your revised manuscript.

Kind regards,

Abhishek Kumar Singh, Ph.D.

Academic Editor

PLOS ONE

“These works are part of the research program Perspectief with project number P18-26, which is financed by the Dutch Research Council (NWO). EvdP acknowledges funding from NWO, VIDI 19724.”

Reviewers' comments:

Reviewer's Responses to Questions

**Comments to the Author**

1. Is the manuscript technically sound, and do the data support the conclusions?

Reviewer #1: Yes

Reviewer #2: Yes

2. Has the statistical analysis been performed appropriately and rigorously? 

Reviewer #1: Yes

Reviewer #2: Yes

3. Have the authors made all data underlying the findings in their manuscript fully available?

Reviewer #1: Yes

Reviewer #2: Yes

4. Is the manuscript presented in an intelligible fashion and written in standard English?

Reviewer #1: Yes

Reviewer #2: Yes

5. Review Comments to the Author

Reviewer #1: The study effectively demonstrates the viability of employing Polaxamer-188 as a substitute for the utilization of BSA or Tween20 as wetting agents in EV determinations via MRPS. I would like to outline certain recommendations to be integrated into the manuscript prior to its publication:

EV test sample characterization by FCM.

The authors employ FCM as a technique for the characterization of EVs, given that it does not require the use of wetting agents and is therefore suitable for assessing the effect of these agents on EVs. As they appropriately clarify, the size detection limit of this technique is approximately 145-165 nm. However, in the Results section, they mention that almost all of the particles in the sample are EVs, when they are only observing those larger than 165 nm using this technique. In fact, in Figure 2A, for particles smaller than 280 nm, there appears to be a decrease in the level of CD235a labeling. While this is more clearly explained in the discussion, it would be useful to mention it in the Results section. It is not evident from this figure whether the data points correspond to a single experiment or are the result of experimental replicates. The same Figure illustrates the effect of Triton X-100 lysis on the particles, indicating that 98% of them were susceptible. However, this seems to hold true for particles larger than 400 nm, but not for the smaller ones. I suggest clarifying how the percentage of lysis is determined in this section.

Effect of wetting agents on EV samples determined by FCM.

In Figure 3, the effect of Tween20 on the particles is evident, but it is not specified whether the results are experimental replicates or single measurements.

Effect of Poloxamer-188 on MRPS determination.

The authors demonstrate a narrower distribution of transit time values when using polaxomer-188 compared to BSA or Tween20, indicating a higher degree of pore wetting (Figure 4). It would be useful to include a measure of dispersion for the values obtained in both cases as supplementary information. Additionally, I suggest presenting the EV concentration values obtained with Tween20 in parallel, in order to showcase the impact of the wetting agent on the final EV concentration determined by MRPS, according to the agent used. This presentation could also reveal whether there is an increase in the concentration of EVs smaller than 200 nm that corresponds to the findings suggested in the FCM assays. Such findings would support the hypothesis of lysis or micelle formation of Tween20, detectable within this size range.

Minor Observations

Figure 1A C: The figure caption mentions that the graph corresponds to Run 2, while the text states that it corresponds to Run 1.

Figures S1.1 and S2.1: Please label panels A and B for clarity.

Reviewer #2: It was the aim of the authors to improve the method microfluidic resistive pulse sensing (MRPS) for measuring of concentration and size of extracellular vesicles by the choice of the wetting compound for the filter that is the part of the measuring device. They found that the previously suggested compound Tween-20 considerably affects the EVs while Poloxamer-188 turned out to be less aggressive and therefore more appropriate. I suggest that the authors consider the comments below:

Line 54: The authors state: »By counting the number of particles and deriving the sample volume, MRPS estimates the concentration of particles.” Please explain what you mean by “Deriving the sample volume”. Figure 1 is a distribution of the number density of particles over the size. To understand this diagram it is necessary to understand how the data are obtained from the measurements. Therefore more explanation and an illustration to support the explanation would be welcome.

The pre-detection filter is an important element of the device. It would therefore be of interest to visualize the filter. Could the authors provide a micrograph showing the size and the shape of the pores?

Lines 64-66. The authors acknowledge the possibility that amphiphile Tween-20 causes lysis of EVs an affects the labelling. However even if the vesicles do not lyse, it could be expected that their interaction with amphiphilic molecules would cause their fragmentation and/or change their size and shape. One would expect that such compounds would be used with intention to strongly modify the membranous systems and affect the measurement of size and concentration.

Extensive research on the effect of amphiphiles on red blood cell vesiculation starting already 50 years ago evidenced strong influence of detergents and other amphiphiles on the membranes. I suggest that the authors point to some of this work (see below some suggestions) and comment on the subject.

1. B. Deuticke, Transformation and restoration of biconcave shape of human erythrocytes induced by amphiphilic agents and changes of ionic environment, Biochim. Biophys. Acta 163, 1968, 494–500.

2. H. Hagerstrand, B. Isomaa, Lipid and protein composition of exovesicles released from human erythrocyte following treatment with amphiphiles, Biochim. Biophys. Acta 1190, 1994, 409–415.

3. Hagerstrand, H. and B. Isomaa. Vesiculation induced by amphiphiles in erythrocytes. Biochim. Biophys. Acta 982, 1989, 179–186.

4. Hagerstrand, H. and B. Isomaa, Morphological characterization of exovesicles and

endovesicles released from human erythrocytes following treatment with amphiphiles.

Biochim. Biophys. Acta 1109, 1992, 117–126.

5. B. Deuticke, R. Grebe, C.W.M. Haest, Action of drugs on the erythrocyte membrane, in: J.R. Harris Ed. , Erythroid Cells, Plenum, New York, 1990, pp. 475–529.

6. Kralj-Iglič V, Pocsfalvi G, Mesarec L, Šuštar V, Hägerstrand H, Iglič A. Minimizing isotropic and deviatoric membrane energy – An unifying formation mechanism of different cellular membrane nanovesicle types. PLoS ONE. 15, 2020, e0244796.

Line 95: The authors state: “This study aims to identify a new wetting agent that leaves EVs intact and effectively reduces the surface tension of the sample for optimal pore wetting.” I see some conceptual problems with this sentence. Could the authors suggest and explain how it would be at least in principle possible to have a surface tension-reducing compound that would leave the membrane in an aqueous medium unaffected. Furthermore, passing through a filter is unlikely to “leave EVs intact” already due to mechanical impact of the material and of the flow. I suggest that the authors rephrase the expression to something like: “This study aims to identify a new agent that causes well controlled effect on EVs for optimal pore wetting.”

Line 100: The authors state: “FCM was selected over MRPS because FCM measurements are virtually independent of the used wetting agents.” I respectfully disagree on this; if the agents modify the physical properties of the membrane (e.g. the bending constant), the flow in FCM could differently affect the samples. It should be born in mind that (in particular erythrocyte) EVs are colloid self-assembly that does not have a fixed identity. However, one could assume that the wetting agents would cause less artefact in measuring with FCM than with MRPS. Please rephrase.

The authors state the lower limit of size assessed by FCM is 145 nm; ultra-resolution FCM may require filtering; were the samples filtered before FCM?

Line 142. The authors claim: “Diluents were filtered (Nuclepore Track-Etch

142 Membrane, 47 mm, 0.05 μm, Whatman) to remove particles smaller than 50 nm.” Please check whether there is a mistake; should it be “to remove particles larger than 50 nm?”

Line 214: Please state the method with which the concentration was measured.

Line 263: The autors state:” The desired wetting agent should fulfill two criteria. First, the wetting agent should leave EVs intact…..” I suggest that the authors decrease the rigor of the expectations. I think that the more appropriate criterion would be “to minimally affect the EVs”.

Line 274: The authors claim: “We chose FCM over MRPS because FCM measurements are

virtually independent of the buffer and also allows reliable size determination”. Please check this sentence. See also the comment above.

Although this is out of scope of the present work, in the future it would be interesting to visualize the samples affected by different wetting compounds before and after MRPS (by SEM and/or cryo-TEM).

6. PLOS authors have the option to publish the peer review history of their article (what does this mean?). If published, this will include your full peer review and any attached files.

Reviewer #1: No

Reviewer #2: No

---

## [Author Response · Author response to Decision Letter 0]

27 Nov 2023

Editor: 

https://journals.plos.org/plosone/s/file?id=wjVg/PLOSOne_formatting_sample_main_body.pdf andhttps://journals.plos.org/plosone/s/file?id=ba62/PLOSOne_formatting_sample_title_authors_affiliations.pdf

Response: 

We would like to express our gratitude to the editor for taking the time to consider our manuscript for publication in PLOS ONE. After carefully reviewing the provided templates, we made the necessary adjustments to ensure that our manuscript complies with PLOS ONE's style requirements.

Editor: 

Response: 

We have thoroughly reviewed the entire manuscript for language usage, spelling, and grammar, and made necessary revisions where required. Rienk Nieuwland was responsible for editing the manuscript.”

Editor: 

3. Thank you for stating the following financial disclosure: “These works are part of the research program Perspectief with project number P18-26, which is financed by the Dutch Research Council (NWO). EvdP acknowledges funding from NWO, VIDI 19724.” Please state what role the funders took in the study. If the funders had no role, please state: "The funders had no role in study design, data collection and analysis, decision to publish, or preparation of the manuscript." If this statement is not correct you must amend it as needed. Please include this amended Role of Funder statement in your cover letter; we will change the online submission form on your behalf. 

Response: 

To clarify the role of the funders, we added the following sentence to the Funding section at page 15 of the revised version of the manuscript: “The funders had no role in the study design, data collection and analysis, decision to publish, or preparation of the manuscript.”

Editor: 

4. In your Data Availability statement, you have not specified where the minimal data set underlying the results described in your manuscript can be found. PLOS defines a study's minimal data set as the underlying data used to reach the conclusions drawn in the manuscript and any additional data required to replicate the reported study findings in their entirety. All PLOS journals require that the minimal data set be made fully available. For more information about our data policy, please see http://journals.plos.org/plosone/s/data-availability. ”

Response: 

We added the following statement to the Data Availability section: “The minimal data set underlying the results can be found at https://doi.org/10.6084/m9.figshare.24298507”

Notification of the authors: 

Authors have identified a marginal error in the manuscript concerning the trigger threshold, that was expressed in terms of the optical diameter of EVs. In the original version of the manuscript, the reported values were 145 nm and 165 nm, where they should consistently have been 145 nm. It is important to note that this correction does not impact the final conclusion of the manuscript. Authors have corrected this mistake in the revised version of the manuscript.

Reviewer 1: 

The study effectively demonstrates the viability of employing Polaxamer-188 as a substitute for the utilization of BSA or Tween20 as wetting agents in EV determinations via MRPS. I would like to outline certain recommendations to be integrated into the manuscript prior to its publication: EV test sample characterization by FCM. The authors employ FCM as a technique for the characterization of EVs, given that it does not require the use of wetting agents and is therefore suitable for assessing the effect of these agents on EVs. As they appropriately clarify, the size detection limit of this technique is approximately 145-165 nm. However, in the Results section, they mention that almost all of the particles in the sample are EVs, when they are only observing those larger than 165 nm using this technique. In fact, in Fig 2A, for particles smaller than 280 nm, there appears to be a decrease in the level of CD235a labeling. While this is more clearly explained in the discussion, it would be useful to mention it in the Results section. It is not evident from this Fig whether the data points correspond to a single experiment or are the result of experimental replicates. The same Fig illustrates the effect of Triton X-100 lysis on the particles, indicating that 98% of them were susceptible. However, this seems to hold true for particles larger than 400 nm, but not for the smaller ones. I suggest clarifying how the percentage of lysis is determined in this section.

Response: 

Thank you for taking the time to review our manuscript and for providing valuable feedback. We appreciate your thoughtful comments, which we hope will contribute to the improvement of our work. In the following list, we will systematically address each of your comments.

 We acknowledge the need to revise the Results section to clarify that the majority of particles with a diameter exceeding 145 nm in the EV test sample are indeed EVs. In consideration of this, we made the following revisions at Page 10 of the original manuscript: 

 We replaced the text “In total 71% of the particles >165 nm and 96% of the particles >285 nm in the test sample exceeded the fluorescence gate of 118 molecules of equivalent soluble fluorophores (MESF) PE, thereby indicating that the majority of particles in the test sample are indeed EVs.” With “In addition, in total 68% of the particles >145 nm and 96% of the particles >285 nm in the EV test sample are stained with CD235a, and therefore likely EVs. ”

 We also replaced “confirming that the majority of particles are EVs.” with “indicating that most particles >145 nm are EVs.” 

 In addition, we replaced the text “Also this result confirms that the majority of particles in the developed test sample are EVs. Based on the findings in Fig 2, we assume that all particles in the test sample are EVs during the follow-up experiments without staining.” with “This RI is typical for EVs within this size range and thereby further confirms that the majority of measured particles in the developed test sample by FCM are EVs [4,18,19]. Based on the findings in Fig 2, we assume that all particles with sizes >145 nm in the test sample are EVs.”

 EVs are heterogeneous particles and detectors of flow cytometers have limited sensitivity. Therefore, only a part of the light scattering distribution and fluorescence distribution of EVs exceeds the background noise. In terms of EV diameter, the trigger threshold, which is placed right above the background noise of the scatter detector, corresponds to 145 nm, whereas the applied fluorescent gate, which is placed right above the background noise of the fluorescence detector, corresponds to mainly particles with a diameter between 145 nm to 280 nm, as shown in Fig. 2B of the revised manuscript. For clarity, we have added the following explanation to Page 10 of the revised manuscript: “In Fig. 2B, we highlight the impact of the fluorescence gate at 118 molecules of equivalent soluble fluorophores (MESF), which not only excludes background noise but also stained events with a fluorescence intensity below 118 MESF. Notably, the data points with the highest density (shown in yellow) and a fluorescence intensity of 118 MESF correspond to particles with an upper size of approximately 285 nm. Consequently, for sizes <285 nm, the fluorescence gate results in a reduced percentage of measured stained particles compared to the total number of particles. This observation shows that the actual percentage of CD235a-PE+ particles exceeds 68%, confirming that the majority of particles >145 nm in the test sample are indeed EVs.”

 The data points in Fig 2 of the original manuscript are derived from a single experiment and we understand that might raise questions from the referee. Therefore, we have conducted a replicate measurement. These data have been added to S3 Fig 1 of the supplementary materials and the results are similar to the original results.

 To clarify how the percentage of lysis is determined, we added the following explanation to Page 10 of the revised manuscript: “The total concentration of detected particles in the EV test sample is 2.0E10 〖ml〗^(-1). After the addition of Triton, the measured concentration decreased to 7.1E8 〖ml〗^(-1), which is similar to the concentration of particles measured in Triton (supplemental materials). This observation implies that 97% of all particles were susceptible to detergent lysis, indicating that most particles >145 nm are EVs.”

 To further address the concern of the reviewer regarding the detergent lysis of EVs, we examined the size distribution of particles in the (267-fold diluted) EV test sample during the flow cytometry measurements in the presence of Triton. These data have been added to S3 Fig 3 of the supplementary materials. The size distribution of particles in the EV test sample in the presence of Triton resembles the size distribution of particles found in a pure Triton solution. This observation verifies that the majority of the particles in the EV test sample that remain after detergent lysis are particles in Triton, most likely micelles. 

Reviewer 1: 

Effect of wetting agents on EV samples determined by FCM. In Fig 3, the effect of Tween20 on the particles is evident, but it is not specified whether the results are experimental replicates or single measurements.

Response: 

 The data points originally presented in Fig 3 of the manuscript were derived from a single experiment. However, to make our findings more trustworthy, we conducted two independent replicates specifically examining the impact of wetting agents, including Tween-20, on EVs. The result of these replicates are similar to our findings in Fig 3 of the original manuscript and these results are added to S3 Fig 2 of the supplementary material.

Reviewer 1: 

Effect of Poloxamer-188 on MRPS determination. The authors demonstrate a narrower distribution of transit time values when using polaxomer-188 compared to BSA or Tween20, indicating a higher degree of pore wetting (Fig 4). It would be useful to include a measure of dispersion for the values obtained in both cases as supplementary information. 

Response: 

 We totally agree with the reviewer that data should be quantified. For a measure of dispersion of the measured transit time values, we used the interquartile range (IQR), as described at Page 11 of the original manuscript: “The IQR of the transit time of particles through the pore is 11 μs for Poloxamer, 56 μs for BSA and 16 μs for Tween.”. We also added “The interquartile range of the transit time of particles through the pore is 11 μs for Poloxamer-188, 56 μs for BSA and 16 μs for Tween-20.”to the caption of Fig 4 of the revised manuscript at Page 13.

Reviewer 1: 

Additionally, I suggest presenting the EV concentration values obtained with Tween20 in parallel, in order to showcase the impact of the wetting agent on the final EV concentration determined by MRPS, according to the agent used. This presentation could also reveal whether there is an increase in the concentration of EVs smaller than 200 nm that corresponds to the findings suggested in the FCM assays. Such findings would support the hypothesis of lysis or micelle formation of Tween20, detectable within this size range. 

Response: 

To address reviewer concern, we added a plot showing the concentration and size distribution of particles in the EV test sample in the presence of BSA, Poloxamer and Tween to Fig 4B of the revised manuscript. To describe the data, we also added the following paragraph to Page 12 of the revised version of the manuscript: “Fig 4B shows the concentration versus size of the EVs in the EV test sample diluted in DPBS containing BSA, Poloxamer, and Tween as measured by MRPS. The concentration of particles in the presence of Tween is 24% higher compared to BSA and Poloxamer. Within the size range of 145 nm to 300 nm, this result aligns with our FCM findings and confirms that Tween affects MRPS measurements of EVs.”

Minor Observations

Reviewer 1: 

Fig 1A C: The Fig caption mentions that the graph corresponds to Run 2, while the text states that it corresponds to Run 1. 

Response: 

 We appreciate your observation regarding the inconsistency between the caption and the text in Fig 1. We made the necessary correction to ensure consistency.

Reviewer 1: 

Figs S1.1 and S2.1: Please label panels A and B for clarity. 

Response: 

 We addressed this issue by adding labels to panels A and B to S1 and S2 Figs for clarity. 

Reviewer 2: 

It was the aim of the authors to improve the method microfluidic resistive pulse sensing (MRPS) for measuring of concentration and size of extracellular vesicles by the choice of the wetting compound for the filter that is the part of the measuring device. They found that the previously suggested compound Tween-20 considerably affects the EVs while Poloxamer-188 turned out to be less aggressive and therefore more appropriate. I suggest that the authors consider the comments below:

Line 54: The authors state: »By counting the number of particles and deriving the sample volume, MRPS estimates the concentration of particles.” Please explain what you mean by “Deriving the sample volume”. Fig 1 is a distribution of the number density of particles over the size. To understand this diagram it is necessary to understand how the data are obtained from the measurements. Therefore more explanation and an illustration to support the explanation would be welcome. 

Response: 

We appreciate your time and effort in reviewing our manuscript and offering valuable feedback. In the following we will answer your comments. 

 We explained the derivation of sample volume in MRPS measurements in the Introduction section of the original manuscript at Page 4: “When a particle passes through the pore, the electrical resistivity of the pore changes due to the difference in conductivity between the particle and the ambient fluid. This change is observed as a pulse of the measured voltage. The amplitude of this pulse is proportional to the particle volume. In addition, the width of the pulse, which is the transit time of the particle through the pore, inversely relates to the flow rate and together with the measurement time provides information about the measured sample volume.” To address the concern of the reviewer, we also added the following explanation to the Method section of the revised manuscript at page 8: “The nCS1 viewer software assumes that the measured sample volume is inversely related to the average transit time of particles passing through the pore, and assesses the sample volume through a calibration.”

 Below, we formulated the process of deriving the sample volume in MRPS measurements: V = F / T 

where V is measured volume, F is calibration factor determined by manufacturer, T is average transit time.

Reviewer 2: 

The pre-detection filter is an important element of the device. It would therefore be of interest to visualize the filter. Could the authors provide a micrograph showing the size and the shape of the pores? 

Response: 

Unfortunately, the exact specifications of the MRPS pre-detection filter are proprietary information of Spectradyne and not available for publication. However, we understand the importance of providing insight into the design of the filter. To address this, we referred readers to a publication by Fraikin et al. (2011) at Page 4 of the revised manuscript: “For MRPS to work correctly, the sample must wet the pre-detection filter, the electrodes and the pore of the microfluidic chip, as depicted in Fig 1 of the manuscript by Fraikin et al. [6].” This reference should help readers gain a general idea of the nanopore structure used in MRPS cartridges.

Reviewer 2: 

Lines 64-66. The authors acknowledge the possibility that amphiphile Tween-20 causes lysis of EVs an affects the labelling. However even if the vesicles do not lyse, it could be expected that their interaction with amphiphilic molecules would cause their fragmentation and/or change their size and shape. One would expect that such compounds would be used with intention to strongly modify the membranous systems and affect the measurement of size and concentration.

Extensive research on the effect of amphiphiles on red blood cell vesiculation starting already 50 years ago evidenced strong influence of detergents and other amphiphiles on the membranes. I suggest that the authors point to some of this work (see below some suggestions) and comment on the subject.

1. B. Deuticke, Transformation and restoration of biconcave shape of human erythrocytes induced by amphiphilic agents and changes of ionic environment, Biochim. Biophys. Acta 163, 1968, 494–500.

2. H. Hagerstrand, B. Isomaa, Lipid and protein composition of exovesicles released from human erythrocyte following treatment with amphiphiles, Biochim. Biophys. Acta 1190, 1994, 409–415.

3. Hagerstrand, H. and B. Isomaa. Vesiculation induced by amphiphiles in erythrocytes. Biochim. Biophys. Acta 982, 1989, 179–186.

4. Hagerstrand, H. and B. Isomaa, Morphological characterization of exovesicles and

endovesicles released from human erythrocytes following treatment with amphiphiles.

Biochim. Biophys. Acta 1109, 1992, 117–126.

5. B. Deuticke, R. Grebe, C.W.M. Haest, Action of drugs on the erythrocyte membrane, in: J.R. Harris Ed. , Erythroid Cells, Plenum, New York, 1990, pp. 475–529.

6. Kralj-Iglič V, Pocsfalvi G, Mesarec L, Šuštar V, Hägerstrand H, Iglič A. Minimizing isotropic and deviatoric membrane energy – An unifying formation mechanism of different cellular membrane nanovesicle types. PLoS ONE. 15, 2020, e0244796. 

Response: 

 We appreciate the insightful suggestion to cite previous research on the influence of amphiphiles on red blood cells. While our data confirm that Tween-20 affects the size distribution and concentration of EVs, we have substantiated our findings by citing the aforementioned articles. We have added the following sentence to page 4 of the revised manuscript: “However, Tween-20 destabilizes the membrane and morphology of erythrocytes [8–12] and might therefore also affect the membrane of EVs.”

Reviewer 2: 

Line 95: The authors state: “This study aims to identify a new wetting agent that leaves EVs intact and effectively reduces the surface tension of the sample for optimal pore wetting.” I see some conceptual problems with this sentence. Could the authors suggest and explain how it would be at least in principle possible to have a surface tension-reducing compound that would leave the membrane in an aqueous medium unaffected. Furthermore, passing through a filter is unlikely to “leave EVs intact” already due to mechanical impact of the material and of the flow. I suggest that the authors rephrase the expression to something like: “This study aims to identify a new agent that causes well controlled effect on EVs for optimal pore wetting.

Response: 

 The comment raises valid points. We acknowledge the conceptual challenges in the original statement. As suggested, we replaced the text at page 5 of the original manuscript “This study aims to identify a new wetting agent that leaves EVs intact and effectively reduces the surface tension of the sample for optimal pore wetting.” with “This study aims to identify a wetting agent that lowers the surface tension of water to achieve optimal pore wetting, while exerting well-controlled effects on EVs. The hallmark of EVs is their phospholipid membrane [14], which require a surface tension to maintain structural integrity. However, the phospholipid membranes of EVs may contain cross-linking proteins that increases their rigidity in the presence of wetting agents [15]. Here we will explore the potential of two non-ionic wetting agents, Poloxamer-188 and Tween-20 for measuring EVs with MRPS.”

Reviewer 2: 

Line 100: The authors state: “FCM was selected over MRPS because FCM measurements are virtually independent of the used wetting agents.” I respectfully disagree on this; if the agents modify the physical properties of the membrane (e.g. the bending constant), the flow in FCM could differently affect the samples. It should be born in mind that (in particular erythrocyte) EVs are colloid self-assembly that does not have a fixed identity. However, one could assume that the wetting agents would cause less artefact in measuring with FCM than with MRPS. Please rephrase.

The authors state the lower limit of size assessed by FCM is 145 nm; ultra-resolution FCM may require filtering; were the samples filtered before FCM? 

Response: 

 To address reviewers concern, we replaced the text at page 5 of the original manuscript “FCM was selected over MRPS because FCM measurements are virtually independent of the used wetting agents.” with “FCM was selected over MRPS because FCM measurements less dependent on the used wetting agents.” It is important to consider that, while hydrodynamic focusing in FCM might potentially affect particle shape, our particles are close to the Rayleigh scattering regime where light scattering measurements are independent of the shape. Additionally, the size distribution of EVs measured by FCM in the presence of Poloxamer-188 is similar to that in our positive controls (DPBS and DPBS with BSA). Hence, any deformation effect caused by hydrodynamic focusing with Poloxamer-188 is likely similar to those without detergent or in the presence of BSA. Furthermore, using different wetting agents for MRPS measurements might necessitate additional calibration, making FCM a more favorable choice. 

 In this study, we did not subject the samples to filtration before FCM analysis. The lower limit of detection by FCM is 145 nm (for particles with a core refractive index of 1.38, a shell refractive index of 1.48 and a shell thickness of 6 nm) due to the sensitivity constraints of light-scattering detectors and the applied trigger threshold.

Reviewer 2: 

Line 142. The authors claim: “Diluents were filtered (Nuclepore Track-Etch

142 Membrane, 47 mm, 0.05 μm, Whatman) to remove particles smaller than 50 nm.” Please check whether there is a mistake; should it be “to remove particles larger than 50 nm?” 

Response: 

 We appreciate your keen observation, there was a mistake in the text. We made the necessary correction in the manuscript. 

Reviewer 2: 

Line 214: Please state the method with which the concentration was measured. 

Response: 

 Thank you for bringing this to our attention. We used FCM to measure the concentration and particle size distribution of the EV test sample, and we added this method to the revised manuscript for clarity. 

Reviewer 2: 

Line 263: The autors state:” The desired wetting agent should fulfill two criteria. First, the wetting agent should leave EVs intact…..” I suggest that the authors decrease the rigor of the expectations. I think that the more appropriate criterion would be “to minimally affect the EVs”. 

Response: 

 Thank you for your suggestion. At page 13 we replaced “leave EVs intact” with “minimally affect the EVs”.

Reviewer 2: 

Line 274: The authors claim: “We chose FCM over MRPS because FCM measurements are

virtually independent of the buffer and also allows reliable size determination”. Please check this sentence. See also the comment above. 

Response: 

 This is a valid point. We replaced the text at page 13 of the original manuscript “We chose FCM over MRPS because FCM measurements are virtually independent of the buffer and also allows reliable size determination” with “We chose FCM over MRPS because FCM measurements are less susceptible to wetting agents than MRPS and FCM also allows reliable size determination [20].”

---

## [Editor Report · Decision Letter 1]

30 Nov 2023

Poloxamer-188 as a wetting agent for microfluidic resistive pulse sensing measurements of extracellular vesicles

PONE-D-23-19466R1

Dear Dr. Shahsavari,

We’re pleased to inform you that your manuscript has been judged scientifically suitable for publication and will be formally accepted for publication once it meets all outstanding technical requirements.

Kind regards,

Abhishek Kumar Singh, Ph.D.

Academic Editor

PLOS ONE

---

## [Editor Report · Acceptance letter]

18 Dec 2023

PONE-D-23-19466R1 

PLOS ONE

Dear Dr. Shahsavari, 

I'm pleased to inform you that your manuscript has been deemed suitable for publication in PLOS ONE. Congratulations! Your manuscript is now being handed over to our production team.

Kind regards, 

on behalf of

Dr. Abhishek Kumar Singh 

Academic Editor

PLOS ONE